# Characterising the Area Under the Curve Loss Function Landscape

## Abstract

One of the most common metrics to evaluate neural network classifiers is the area under the receiver operating characteristic curve (AUC). However, optimisation of the AUC as the loss function during network training is not a standard procedure. Here we compare minimising the cross-entropy (CE) loss and optimising the AUC directly. In particular, we analyse the loss function landscape (LFL) of approximate AUC (appAUC) loss functions to discover the organisation of this solution space. We discuss various surrogates for AUC approximation and show their differences. We find that the characteristics of the appAUC landscape are significantly different from the CE landscape. The approximate AUC loss function improves testing AUC, and the appAUC landscape has substantially more minima, but these minima are less robust, with larger average Hessian eigenvalues. We provide a theoretical foundation to explain these results. To generalise our results, we lastly provide an overview of how the LFL can help to guide loss function analysis and selection.

## 1 Introduction

The area under the curve of the receiver operating characteristic curve (AUC) is a commonly used method to evaluate the accuracy and reliability of a neural network classifier. However, for mathematical reasons, the AUC cannot be used as as the loss function to be minimised during neural network training (Menon & Elkan, 2011). Instead, other functions such as cross-entropy are commonly employed. The stepwise nature of the AUC function is the reason that it is non-differentiable and hence cannot simply be optimised. However, the AUC can be approximated using surrogate losses such as the sigmoid function. This approach can lead to a formulation equivalent to the soft-AUC described by Calders & Jaroszewicz (2007), which is referred to here as appAUC. We find it intuitive to optimise a function that it as close as possible to the one used to evaluate the model. Our main contributions in this paper are therefore:

- To understand the use of different approximation to the AUC as the loss function employed in training of a neural network
- To explore the organisation of the appAUC landscape and compare it to a 'standard' cross-entropy landscape
- A theoretical foundation of the differences between appAUC and cross-entropy landscapes
- To outline how a loss function landscape analysis can be useful for loss function selection

To better understand the advantages and disadvantages of the approximated AUC loss function, we study the functional space, commonly referred to as LFL, using tools from the theoretical study of energy landscapes in molecular and condensed matter systems (Wales, 2003). The usefulness of the energy landscape approach has previously been demonstrated in the context of neural network LFLs (Ballard et al., 2017). We will employ methods from this approach to gain insights about geometric features of the LFL, including the number of minima, their curvatures, and their connectivity. By repeatedly surveying large parts of the LFL, we are, with high probability, able to find the true global minimum. Additionally, due to a Metropolis criterion in the global optimisation approach described below, we do not get stuck in a local minimum, and explore the full LFL, hence learning more about the functional surface. Instead of a single minimum, we aim to find a large number of minima that,

together with transition states, provide a faithful coarse-grained representation of the loss function landscape. We believe that this approach will yield valuable insights into the use of appAUC as a loss function in neural networks.

Various interesting questions arise from the use of an appAUC loss function. Besides a comparison of properties between landscapes, and the effects of hyperparameter changes, we are especially interested in the differences between appAUC and CE landscapes. Both loss functions, for the same neural network architecture, address the learning problem of finding a mapping $f$ from input data to class label. Does this common foundation imply that minima of the CE loss function are also minima of the appAUC function, or are they at least very similar to each other? We will show below that this condition does not hold, and explain why. Furthermore, to the best of our knowledge, there has been no previous research into the functional properties of AUC surrogates. We believe that quantifying inherent, geometric properties of loss functions will provide a more fundamental understanding of the applicability of particular loss functions to distinct machine learning problems.

## 1.1 RELATED WORK

This contribution lies at the intersection of three different areas of research: general understanding and study of loss functions, the appAUC loss function in particular, and the study of loss function landscapes to understand neural networks. Optimising the AUC as a loss function has been considered before (Cortes & Mohri, 2004). It has been shown that testing AUC is improved when a loss function is chosen that is closer to the true AUC function (Yan et al., 2003). Nonetheless, optimising loss functions that approximate the AUC is rarely considered. One reason for this situation may be that the computational complexity is formally $\mathcal{O}(N^2)$, or specifically in the binary classification case employed here, $\mathcal{O}(N_P N_N)$ where $N_P$ are the positive data points and $N_N$ the negative ones. Other reasons include the non-convexity and perceived complexity of the appAUC landscape, and importantly the fact that it has zero derivatives almost everywhere (Ghanbari & Scheinberg, 2018). The loss function is one of the most critical choices in the design of a neural network, because it is the element that underlies the entire learning procedure. At first glance, it may appear that all loss functions solve the same problem for a given neural network, which led Rosasco et al. (2004) to question the practical differences between alternative loss functions. Yet, it is widely accepted that different loss functions allow us to optimise for specific properties (Janocha & Czarnecki, 2017). By studying the appAUC LFL, we can quantitatively address these important questions.

The topology of the loss function has been a topic of interest for over 20 years (Hochreiter & Schmidhuber, 1997). Understanding the loss function for a given neural network can help to establish the fundamental interpretation of the 'black-box', as such networks are often described (Li et al., 2017). However, a key problem with studying the LFL is the large associated computational cost. Unlike the standard machine learning approach, where only a single minimum is located, the LFL approach attempts to sample a representative set of minima, which may often exceed 10,000 solutions (Ballard et al., 2017). Computing the true number of minima for a loss function would require enhanced sampling techniques (Wales, 2013; Martiniani et al., 2016).

Recently, others have looked into the LFL of overparameterised neural networks (Cooper, 2018) and shown that various properties can be exploited to improve accuracy and, importantly, robustness of neural networks (Baldassi et al., 2020; Chaudhari et al., 2019). Insights from the LFL have also been used to understand initialisation methods and their relative success (Fort & Scherlis, 2019). However, to the best of our knowledge, there has been no research so far into interpreting individual loss functions from the underlying LFL. In this work, we aim to address this knowledge gap, and show that studying the LFL can provide important insights into our understanding and design of neural networks and the associated loss functions.

## 2 METHODS

### 2.1 NEURAL NETWORK

For this initial survey, we consider neural networks with a single hidden layer. We denote the set of data points as $\mathcal{D} = (\mathbf{X}, \mathbf{c})$, containing $N := |\mathcal{D}|$ elements. For a given classification problem with $C$ classes, a data point $d \in \mathcal{D}$ has input features $\mathbf{x}^d$, and a known output class, $c^d$. We use a nonlinear $\tanh$ activation function and convert output values $y_i, i \in C$ into softmax probabilities by

$p_i(\mathbf{W}; \mathbf{x}^d) = \exp[y_i(\mathbf{W}; \mathbf{x}^d)] / \sum_{j=1}^{C} \exp[y_j(\mathbf{W}; \mathbf{x}^d)]$, where $\mathbf{W}$ denotes the weight vector containing all weights of the neural network. As a reference to compare with our appAUC loss function, we use a cross-entropy (CE) loss function, defined as:

$$\text{CE}(\mathbf{W}; \mathbf{X}) = -\frac{1}{N} \sum_{d=1}^{N} \ln(p_{c^d}(\mathbf{W}; \mathbf{x}^d)) + \lambda \mathbf{W}^2. \tag{1}$$

The $\lambda \mathbf{W}^2$ in equation 1 represents an L2 regularisation term, which eliminates zero Hessian Eigenvalues and counteracts overfitting. We fix $\lambda = 10^{-5}$ for a fair comparison between loss functions.

## 2.2 AREA UNDER THE CURVE

To evaluate a model such as the one described above, it is standard practice to consider the receiver operating characteristic (ROC) and calculate the area under the curve (AUC) (Hastie et al., 2009). For a given classifier, the ROC is a plot of the true positive ratio, $T(P)$, against the false positive ratio, $F(P)$. These quantities are defined by

$$T(\mathbf{W}; \mathbf{X}, P) = \frac{\sum_{d=1}^{N} \delta(c^d - 1) \Theta(p_1(\mathbf{W}; \mathbf{x}^d) - P)}{\sum_{d=1}^{N} \delta(c^d - 1)}, \tag{2}$$

$$F(\mathbf{W}; \mathbf{X}, P) = \frac{\sum_{d=1}^{N} (1 - \delta(c^d - 1)) \Theta(p_1(\mathbf{W}; \mathbf{x}^d) - P)}{\sum_{d=1}^{N} 1 - \delta(c^d - 1)}, \tag{3}$$

where $\delta(c^d - 1)$ is the Dirac delta function, and $\Theta(p_1 - P)$ is the Heaviside step function, defined as

$$\delta(c^d - 1) = \begin{cases} 1 & \text{if } c^d = 1, \\ 0 & \text{if } c^d \neq 1, \end{cases} \qquad \Theta(p_1 - P) = \begin{cases} 1 & \text{if } p_1 \geq P, \\ 0 & \text{if } p_1 < P, \end{cases} \tag{4}$$

and $P$ is a parameter, which acts as a cutoff probability for the neural network to predict that a given data point belongs to class 1 ($p_1$). Note that the choice of class 1 as a positive reading is arbitrary, and for a multi-class system, any class may be chosen.

The ROC curve is defined by $T$ and $F$ as $P$ varies from 0 to 1. For a perfect classifier, the ROC would simply be a horizontal line at $T = 1$ (i.e. $\forall P \neq 0, \ T(P) = 1$), and the area under the curve would then be 1. The AUC is therefore a measure of how close the model is to a perfect classifier. Formally, the AUC as a function of network weights $\mathbf{W}$, parameterised by the data $\mathbf{X}$, is given by

$$\text{AUC} = \int_0^1 T(\mathbf{W}; \mathbf{X}, P) \, \mathrm{d}F(\mathbf{W}; \mathbf{X}, P) = \frac{1}{N_P N_N} \sum_p \sum_n \Theta(p_1(\mathbf{W}; \mathbf{x}^p) - p_1(\mathbf{W}; \mathbf{x}^n)) \tag{5}$$

where p labels positive data points (class 1), and n labels negative data points (not in class 1).

## 2.3 APPROXIMATED AUC: APPAUC LOSS FUNCTION

Most optimisation routines benefit from analytical derivatives of the function to be optimised. The function defined in equation 1 has smooth analytical derivatives, but equation 5 does not, because of the discontinuous step function. However, in a similar way to other approaches (e.g. (Calders & Jaroszewicz, 2007)), one can replace the discontinuous $\Theta$ function with an approximate, smooth, analytical surrogate function, which can then be differentiated and optimised. We write

$$\text{AUC}(\mathbf{W}; \mathbf{X}) \approx A(\mathbf{W}; \mathbf{X}) \equiv \frac{1}{N_P N_N} \sum_p \sum_n \frac{1}{1 + \exp(-\beta(p_1(\mathbf{W}; \mathbf{x}^p) - p_1(\mathbf{W}; \mathbf{x}^n)))}, \tag{6}$$

where the Heaviside step function has been replaced with a smooth sigmoid function:

$$\Theta(z) \rightarrow \sigma(z) \equiv \frac{1}{1 + \exp(-\beta z)}, \tag{7}$$

with parameter $\beta$ that is discussed in detail in the Appendix. An important consideration for surrogate AUC loss functions is that they do not change the optimal solution when replacing the step function. Such surrogates are referred to as AUC-consistent (Agarwal, 2014). Charoenphakdee et al. (2019) show that sigmoid is an AUC-consistent surrogate. The appAUC loss function in equation 6 is now differentiable and can hence be used for optimisation, where we minimise the negative form of equation 6 such that a minimum (i.e. lower loss) is a good solution of higher AUC. For reference, we have included the analytical first and second derivatives for Equation 6 in the Appendix.

### 2.3.1 OTHER SURROGATE LOSS FUNCTIONS

There exist many possible ways to approximate the AUC. Besides the sigmoid (Eq.7), we also show results for the popular (Gao & Zhou, 2015) hinge $\ell(z) = \max(0, 1 - z)$, exponential $\ell(z) = exp(-z)$, quadratic $\ell(z) = (1 - z)^2$ and tanh surrogates (Fig. 1), where $z$ denotes the difference in probability between true and false class i.e. $P_T - P_N$. Note that for the sigmoid in Figure 1, we use $\beta = 40$ (Appendix). It is important to note that all but the hinge loss are AUC-consistent (Gao & Zhou, 2015; Charoenphakdee et al., 2019). AUC-consistency implies that the optimal solution remains unchanged with the surrogate loss replacing the true AUC. These different losses can all be used as surrogates because they include a pairwise comparison of positive and negative case probabilities as for the AUC.

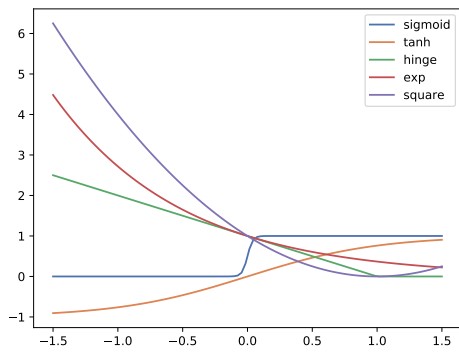

Figure 1: Surrogate loss functions.

## 2.4 OPTIMISATION ROUTINES

Surveying the LFL for a neural network requires methods to locate minima of the landscape and connect them via transition states, defined as index one saddles (Murrell & Laidler, 1968). We perform global optimisation using the basin-hopping method, which uses a Metropolis criterion to avoid getting stuck in local minima (Li & Scheraga, 1987; Wales & Doye, 1997). Further details regarding landscape exploration using basin-hopping are included in the Appendix. We use a modified quasi-Newton L-BFGS minimiser (Nocedal, 1980). Connectivity between minima is defined by transition states using a doubly-nudged (Trygubenko & Wales, 2004a;b) elastic band (Henkelman & Jónsson, 2000; Henkelman et al., 2000) approach and hybrid-eigenvector following (Munro & Wales, 1999; Zeng et al., 2014). These methods are implemented in the GMIN (Wales, a), OPTIM (Wales, b) and PATHSAMPLE (Wales, c) programs, which are available for use under the GNU GPL.

## 2.5 DISCONNECTIVITY GRAPHS

To visualise the high-dimensional loss function landscape, we employ disconnectivity graphs (Becker & Karplus, 1997; Wales et al., 1998), which provide a characteristic representation of the organisation for a high-dimensional surface. The vertical axis corresponds to increasing loss value, highlighting the barrier heights between local minima, and the horizontal arrangement is chosen to avoid crossings between the branches. In molecular science the vertical axis corresponds to potential or free energy, so the loss value plays the role of the energy in the LFL. Any node at which two or more branches split up can be understood as the minimum energy (loss) required for the system to move between the sets of minima associated with the respective branches. Such points correspond to the energies of transition states, discretised at a regular set of threshold values. Each leaf (tip) of the graph corresponds to a minimum of the LFL; the lower the leaf is, the lower its loss value. The lowest-lying leaf is the global minimum. Disconnectivity graphs provide a concise and intuitive way to visualise complex topologies, and preserve a faithful representation of the barriers on the landscape.

## 2.6 DATASET

The LFL is not only a function of the weights, but also of the data, as shown in equation 6. We will present our analysis of the AUC landscape mostly using the synthetic spiral dataset, introduced by Lang & Witbrock (1988). Spiral data remains one of the most popular synthetic datasets due to its high degree of non-linear separability. We add a small uniform noise term to increase the complexity of the problem and with an 80-20 train-test split. We acknowledge the limitations of applicability of a synthetic dataset and therefore supplement our results using a real-world dataset of fraudulent credit card transactions (Le Borgne & Bontempi, 2004). The dataset is highly imbalanced with only 10% of all datapoints classified as fraudulent transaction.

## 3 RESULTS

In this section, we highlight three distinct points of interest. Firstly, we visualise and outline key differences between the AUC and CE landscapes. Secondly, we present an extended quantitative and geometric analysis of different AUC surrogates. Thirdly, we provide a comparison between minima of the AUC and CE landscape in terms of their classification properties.

### 3.1 LFL PROPERTIES

The loss function landscapes for AUC-approximated and CE loss functions look substantially different (Fig. 2). We performed landscape exploration using PATHSAMPLE until no new minima were found. For computational reasons, we only compare sigmoid appAUC with CE, and show below that for shorter timescales, sigmoid looks similar to the other surrogates. In particular, we find that the CE landscape is much more convex or funnelled than the appAUC landscape, with larger uphill barriers. In molecular science, this structure is associated with self-organising systems where relaxation to the global minimum is relatively efficient (Wales, 2003).

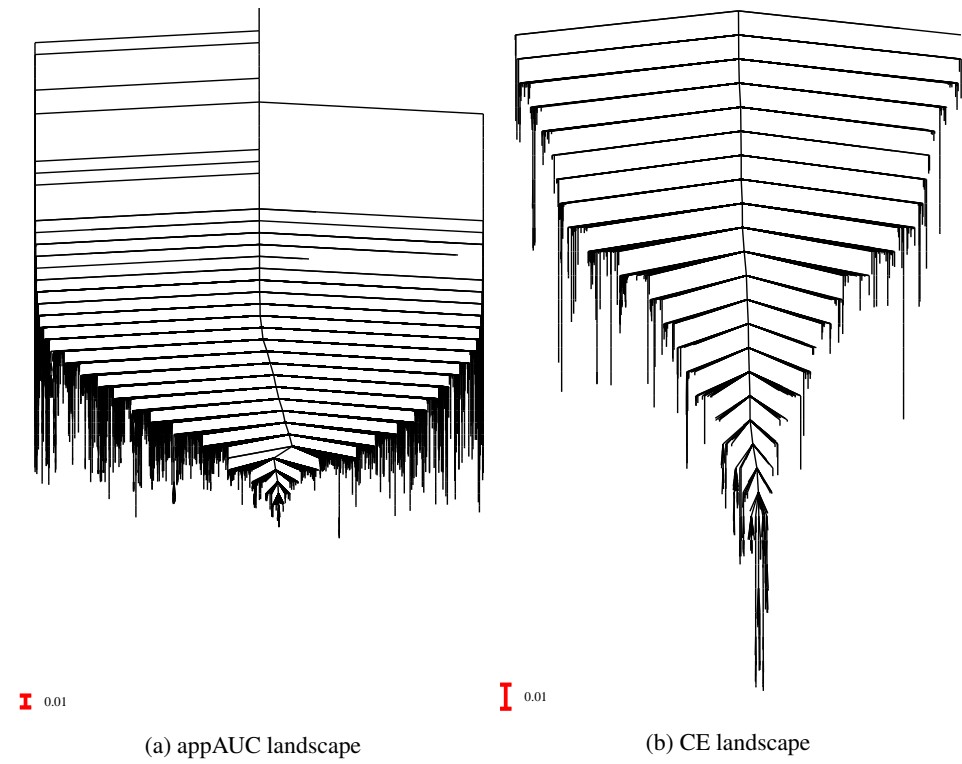

(a) appAUC landscape                   (b) CE landscape

Figure 2: Disconnectivity graphs showing characteristic LFLs for sigmoid and CE. The vertical axes of the landscapes are not directly comparable, as they correspond to different loss functions.

Table 1: Characteristic summary statistics for LFLs after exhaustive landscape exploration.

| | | AUC | | | | LPPHE | |
|---|---|---|---|---|---|---|---|
| Loss function | # minima | best | $\mu$ | $\sigma$ | # transition states | $\mu$ | $\sigma$ |
| Sigmoid | 13,948 | 0.81 | 0.7 | 0.04 | 25,957 | -93 | 7.2 |
| CE | 1,903 | 0.77 | 0.66 | 0.03 | 3,606 | -110 | 4.4 |

In Table 1 we report some summary statistics to characterise the two different LFLs numerically. The last column reports the mean of the log-product of positive Hessian Eigenvalues (LPPHE) for all minima of the landscape. This measure of basin geometry provides insight into the local curvature

and hence the 'flatness' of a minimum (Verpoort et al., 2020) and the volume of the corresponding basin of attraction (Mezey, 1987; Wales, 2003). The smaller the LPPHE, the flatter the local basin. Table 1 shows that the appAUC landscape has substantially more minima and transition states than the CE landscape. Further, the mean LPPHE of all appAUC minima is larger than for the CE minima.

## 3.2 AUC SURROGATES

Table 2 shows a comparison of the different appAUC surrogates (Section 2.3.1) after a week of landscape exploration. We find that these functions behave similarly, with comparable AUCs found and more minima than the CE landscape. The mean LPPHEs vary, the higher values belonging to the two functions that are true approximations to the stepwise AUC, namely sigmoid and tanh. They are all nonetheless larger or equal to the CE LFL in Table 1. The disconnectivity graphs for these alternative appAUC functions are shown in Figure 3. They are all similar to the appAUC landscape in Figure 2 in the sense that they appear very wide (referred to as "glassy" in molecular sciences) and much less single-funnelled than the CE LFL.

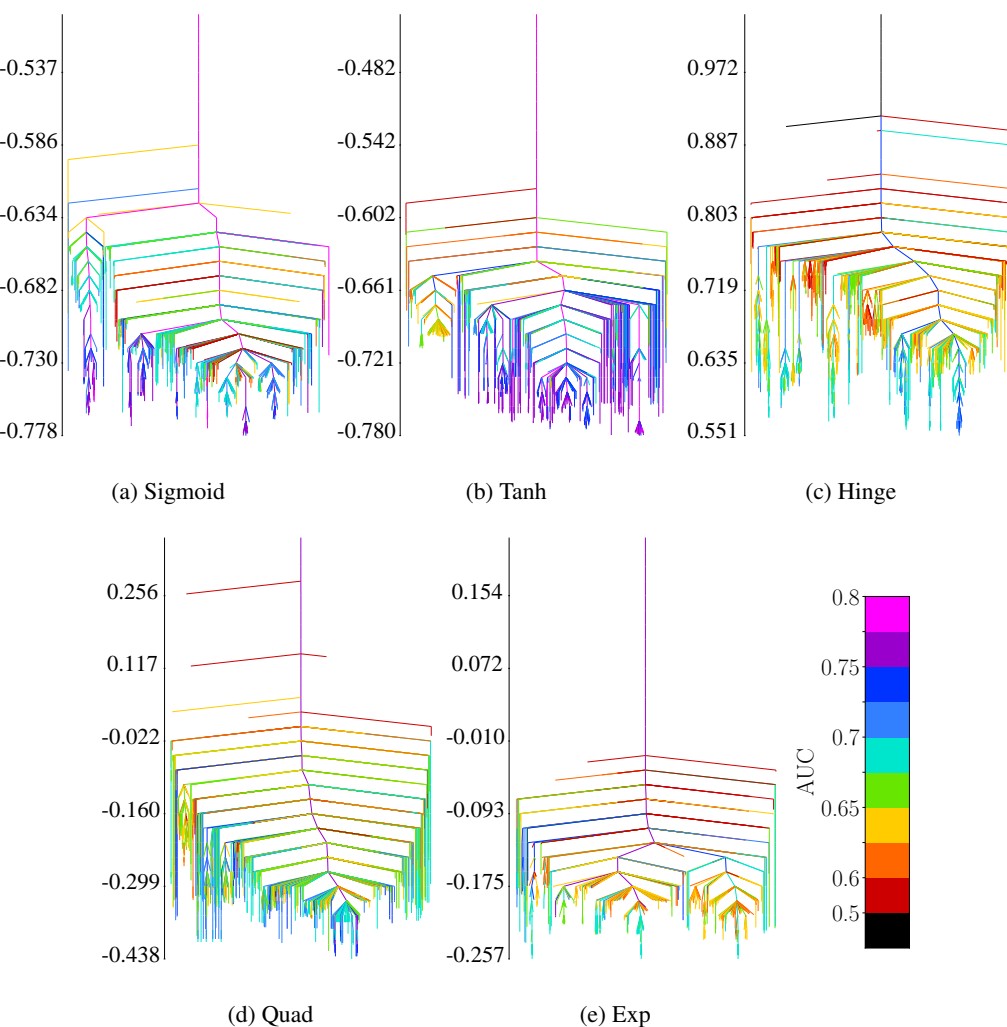

Figure 3: Disconnectivity graphs showing the LFLs for the alternative surrogate AUC loss functions. The minima are coloured by testing AUC. Note that for sigmoid and tanh, the energy/loss value can directly be interpreted as (negative) AUC.

Table 2: Summary statistics for LFLs with alternative appAUC functions and CE as comparison.

| Loss function | # minima | AUC | | | # transition states | LPPHE | |
|---|---|---|---|---|---|---|---|
| | | best | $\mu$ | $\sigma$ | | $\mu$ | $\sigma$ |
| Sigmoid | 4,202 | 0.78 | 0.67 | 0.04 | 4,264 | -99 | 10 |
| Hinge | 6,086 | 0.74 | 0.64 | 0.03 | 5,822 | -109 | 12 |
| Tanh | 3,562 | 0.79 | 0.68 | 0.05 | 3,000 | -94 | 14 |
| Quad | 3,606 | 0.77 | 0.65 | 0.04 | 3,464 | -103 | 11 |
| Exp | 2,563 | 0.77 | 0.64 | 0.03 | 2,515 | -110 | 12 |

### 3.3 LOSS-AUC CORRELATION

Below, we show the correlation between loss value (energy) and AUC for test data for the different surrogate losses (Figure 4). In all five cases, there is a significant negative correlation. Tanh has the most negative Pearson correlation coefficient of -0.83, followed by sigmoid and hinge.

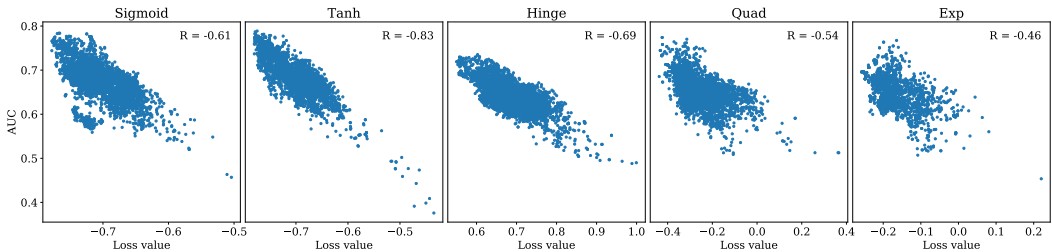

Figure 4: Loss vs AUC correlation for the five different AUC surrogate functions. Minima at lower values of loss generally correspond to higher AUC values.

### 3.4 APPAUC MINIMA IN THE CE LFL

Another point of interest is to understand the relationship between CE loss value and testing AUC for different LFLs. In Figure 5 we use sigmoid to represent appAUC. We see that the negative correlation between CE loss and AUC is much stronger for minima of the CE than for the appAUC landscape. Hence, for most minima of the appAUC landscape, the CE loss values are relatively high, even though some of them may in fact have a higher AUC that the CE minima. This result raises the question of how good a diagnostic the CE loss is for the testing AUC. The insights into the position of appAUC minima within the CE LFL can be combined with the previous results on appAUC minima and the minimum they map to in the CE LFL. Above, we have identified that there exist many appAUC minima that are not minima of the CE LFL and thus have a higher CE loss value, but have comparable testing AUC. In general, while the best appAUC minima have reasonably low

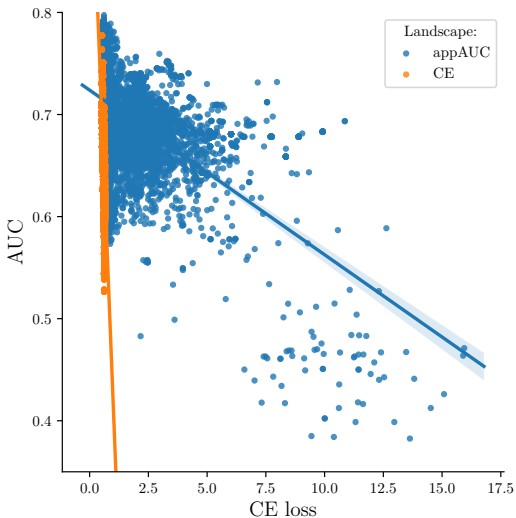

Figure 5: Correlation between CE loss and testing AUC for minima from CE and sigmoid LFLs.

CE loss, the mean loss of 1.2 is nearly twice as high as for CE minima. The median of 0.8 is slightly lower, indicating a few very high loss minima, which lie in the lower right hand corner of Figure 5.

### 3.5 CREDIT CARD DATASET

We perform the same analysis on the imbalanced, real-world credit card dataset. Optimising AUC on such datasets is expected to particularly useful, because the AUC as a measure is invariant to class imbalance. The appAUC landscape has more, better minima, and larger LPPHEs (Table 3). As above, the CE landscape is single funnelled while appAUC landscape is more glassy, similar to figure 2.

Table 3: Summary statistics for credit card fraud LFLs

| | | AUC | | | | LPPHE | |
| Loss function | # minima | best | $\mu$ | $\sigma$ | # transition states | $\mu$ | $\sigma$ |
| --- | --- | --- | --- | --- | --- | --- | --- |
| sigmoid | 3,448 | 0.99 | 0.96 | 0.02 | 2343 | -772 | 38 |
| CE | 1,721 | 0.97 | 0.96 | 0.01 | 1584 | -858 | 9 |

## 4 DISCUSSION

We observe that the appAUC and CE LFLs have rather different characteristics, emphasising the point that different loss functions are truly different. In this section, we discuss and explain these observations, both empirically and theoretically. We explain why an analysis of the LFL can offer unique insights into the understanding of loss functions. Different characteristics of such a function may be important, and studying the LFL can help to choose an appropriate loss function. We also provide a comparison of the different AUC surrogates.

### 4.1 COMPARISON OF LOSS FUNCTION LANDSCAPES

As the disconnectivity graphs in Figure 2 show, the CE landscape is more funnelled, and there are fewer minima. However, not only does the number of minima vary, the LPPHE also indicates that the minima appear rather different. Flatter minima (smaller LPPHE) are sometimes considered advantageous, because they may be more robust to perturbations in the weights (Hochreiter & Schmidhuber, 1997), and therefore potentially also to noise in the training data. The increased robustness is explained by a lower deviation from the loss value of the global minimum for small deviations in training data or model hyperparameters in flatter landscapes where the gradients are small. This feature may constitute a geometrically-inspired argument against using the appAUC landscape if appAUC minima are less robust. The data suggest that on average the CE minima are relatively flat, which may translate to increased robustness to perturbations of the weights.

The location of appAUC minima within the CE LFL raises several new questions. Figure 5 illustrates the weakly negative correlation between CE loss and testing AUC for appAUC minima, much weaker than for CE LFL minima. This result initially seems intuitive, as we are looking at the CE LFL. The fact that the correlation between testing AUC and loss is more strongly negative for points of the CE loss function means that the CE loss works well for the minima in its own LFL. However, there also exist points in the CE LFL that are not minima (but minima of the appAUC) with higher loss and also higher AUC. Hence, under the assumption that appAUC is the more 'accurate' loss function, this analysis highlights some weaknesses of the CE LFL, namely not identifying as many better, correct solutions. A mathematical foundation of the limitations of CE as opposed to appAUC is given in the next section.

### 4.2 THEORETICAL INTERPRETATION

To explain our empirical results, we can provide a theoretical basis for some of the observations made. Notably, the substantially higher number of minima of the appAUC LFL is explained by mathematical properties of the solution space for CE in relation to appAUC loss generally. In fact, the solutions to CE loss are a subset of the solutions to the appAUC loss (Menon & Williamson, 2016). The optimal solution of appAUC is any function that has a strictly monotonic relationship with $P(y = 1|x)$ (Gao & Zhou, 2015). On the other hand, the optimal solution of CE loss is $P(y = 1|x)$. Thus, a space of solutions for appAUC is infinitely larger than CE loss and therefore it should have more minima. The

CE loss with Softmax belongs to the class of 'strictly proper composite' losses (Reid & Williamson, 2010) which estimates $P(y = 1|x)$. In statistics, this is also referred to as 'proper scoring rule' (Buja et al., 2005). Obviously, $P(y = 1|x)$ has a strictly monotonic relationship with itself and hence must be optimal w.r.t. AUC, however this relationship must not hold the other way, optimising AUC must not necessarily be optimal w.r.t. $P(y = 1|x)$. Furthermore, the same line of argument also allows us to explain why appAUC minima are not necessarily close to CE minima in Euclidean space. Minimising appAUC does not give any additional incentive to find solutions of the kind $P(y = 1|x)$ compared to any other solutions which may have a strictly monotonic relationship with $P(y = 1|x)$.

### 4.3 APPAUC VS CE LOSS FUNCTION

The appAUC landscape we have considered has approximately 8 times as many minima as the corresponding CE landscape. The global minimum for appAUC achieves better testing AUC, and the mean AUC is also higher, yet there are also a few especially poor minima in the AUC landscape that are not found for the CE case. For practical applications the existence of solutions that are poor classifiers is probably not a concern, because finding low-lying minima is relatively straightforward. However, computation of the appAUC as done here is of order $\mathcal{O}(N^2)$, where $N$ is the number of data points, which is much more intensive that the $\mathcal{O}(N)$ requirement associated with CE. Additionally, minima of the CE LFL seem to be more robust. In summary, exploiting the appAUC directly is more attractive if computational cost is not a problem, especially if the loss function landscape can be studied extensively. For machine learning applications involving large deep networks, where only a single minimum is identified, and where computational cost is a major concern, the CE landscape may be the best choice. This result strengthens the case for using CE as a 'standard' loss function, while also highlighting its limitations, and illustrating how using different loss functions can be beneficial.

### 4.4 AUC SURROGATES

We observe that the LFLs for all AUC surrogates look roughly similar, and distinctly different to the CE LFL, providing a strong argument for a fundamental geometric difference between these loss functions that has not been shown before. We also observe that the hinge loss has substantially more minima than all other surrogates, yet the testing AUCs are much worse. This may be explained by the fact that hinge is not AUC consistent which means that the hinge functional space does not contain the AUC-optimal solution for this problem. Quad, tanh and sigmoid have broadly similar characteristics with small deviations in the LPPHE while the exponential does slightly worse in terms of AUC but has very flat minima, similar to the CE landscape. Hence, using an exponential AUC surrogate may provide more robust solutions than other surrogates. If the main objective however is to maximise testing AUC, tanh seems to be the best choice, as it not only has the highest AUC, but also the strongest inverse correlation between energy and AUC. Both tanh and sigmoid are directly approximating the stepwise AUC which explains the strongly negative correlations, while both exponential and squared exponential function behave broadly similar as expected.

### 4.5 CONCLUSIONS

Loss functions are essential to guide the learning process in neural networks. We have provided a detailed analysis of the approximate AUC landscape based on global exploration of tractable but realistic examples. In this contribution, we have proposed a new method to understand and select a loss function, based on geometric properties of the LFL. We have shown that LFLs for appAUC and the standard CE LFL are qualitatively and quantitatively different. We have also provided a detailed comparison of relevant LFL features between AUC surrogates and shown that the AUC-inconsistent hinge has much lower AUC than the other surrogates (but more minima) and that the exp surrogate has wide, robust minima, similar to the CE LFL. It is outlined that optimising the AUC improves testing AUC, but CE minima have on average smaller LPPHEs, which may make them more robust. In general, we observe a tradeoff between robustness and testing AUC for all the loss functions. Furthermore, it must be noted that the greater computational cost of optimising the AUC directly is likely to be the main drawback of this approach. A quantitative analysis of geometric features of the LFL such as by-minimum AUC, energy-testing AUC correlation, number of minima (landscape convexity) and LPPHE i.e. catchment basin volumes which is connected to minima robustness allows insights into the strengths and weaknesses of loss functions. Hence, this analysis provides a valuable tool to guide loss function selection for machine learning applications.

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

# 5 APPENDIX

## 5.1 BASIN-HOPPING OPTIMISATION

Surveying the LFL for a neural network requires methods to locate minima of the landscape and connect them via transition states, defined as index one saddles (Murrell & Laidler, 1968). We perform global optimisation using the basin-hopping method, which uses a Metropolis criterion to avoid getting stuck in local minima (Li & Scheraga, 1987; Wales & Doye, 1997). For local minimisation we use a modified quasi-Newton L-BFGS (Limited-memory Broyden–Fletcher–Goldfarb–Shanno) algorithm (Nocedal, 1980). After a local basin of attraction (local minimum) is found via L-BFGS optimisation, a 'basin-hopping' jump is performed to some other point in the LFL. This jump is always accepted if the energy, here loss value, is lower than the current minimum. If the loss value of the new point is higher, it is accepted with probability

$$P \propto \exp\left(-\frac{\Delta \widetilde{E}}{k_\mathrm{B}T}\right) \tag{8}$$

where $\Delta \widetilde{E}$ is the difference in loss value between the current minimum and the new point, $k_\mathrm{B}$ the Boltzmann constant and $T$ a fictitious temperature. Intuitively, if the energy difference between the old and new points is large, the move is less likely to be accepted. A minimum is characterised as such if a RMS norm of gradient vector) convergence criterion at a threshold of $10^{-10}$ is reached.

## 5.2 HYPERPARAMETER ABLATION STUDY

### 5.2.1 SIGMOID

The relevant hyperparameter in the sigmoid function is $\beta$ in the exponent (see Equation 7). For larger $\beta$, the function becomes more stepwise, i.e. a better approximation to the true AUC, with

$$\lim_{\beta \to \infty} \sigma(z; \beta) \to \Theta(z), \tag{9}$$

where again $\sigma(z)$ denotes the sigmoid and $\Theta(z)$ the Heaviside step function. We are interested in the impact of the hyperparameter $\beta$ on the appAUC LFL. Figure 6 shows the effect on the loss function landscape of increasing $\beta$ (on a log scale). Clearly, both the number of minima in the LFL and the AUC of the best minimum increase with $\beta$. The $\Delta$AUC plot in Figure 6 shows the median absolute difference between the approximate AUC value, i.e. the value of the loss function, and the true AUC, for all minima found at a specific $\beta$. This value is inversely proportional to $\beta$ before it plateaus off at a median AUC difference of around 0.04. This median absolute difference is a relevant metric because it shows how closely the appAUC approximates the true AUC value. The rightmost plot in Figure 6 shows a measure of the computational cost associated with optimising the appAUC loss function for different values of $\beta$. The number of optimisation steps for each iteration of the L-BFGS optimiser grows roughly linearly with $\beta$.

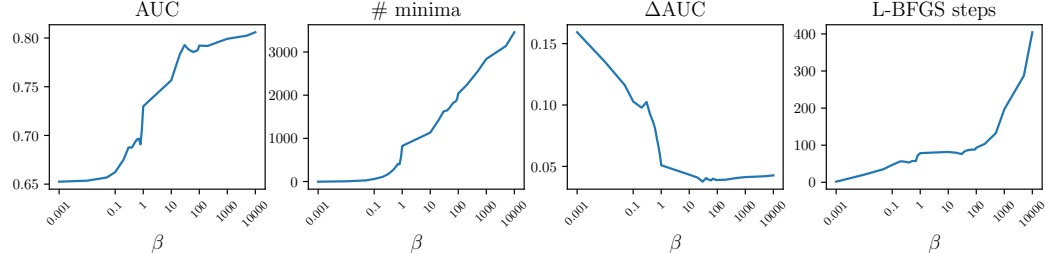

Figure 6: AUC of the best minimum, number of minima of the LFL, difference between appAUC loss value and true AUC, and number of steps per optimiser basin-hopping run for increasing values of hyperparameter $\beta$ in the sigmoid function. The horizontal axes are plotted on a log-scale for visualisation purposes. The results are smoothed using a 2nd-order Savitzky-Golay filter.

The difference between the true AUC and the approximated AUC is inversely proportional to $\beta$, as expected, because for larger $\beta$, the AUC approximation becomes more stepwise and hence more

exact. We find that for $\beta > 50$, the median difference goes to around 0.04, which means it is a reasonably good approximation of the true AUC. We also observe an increase in best AUC, as well as the number of minima of the LFL with increasing $\beta$. All of these observations suggest that larger values of $\beta$ are better, but increased performance comes at an increased computational cost. The reason for this increase is that the number of optimisation steps in the L-BFGS routine increases for larger $\beta$. Most likely, this result can be explained by the flatter appAUC function at large $\beta$ around local minima. In flatter regions, convergence to a true minimum usually requires more optimisation steps because the function is more anharmonic. To conclude, we suggest a value for $\beta$ of around 40. For higher values, the AUC does not improve significantly, while the computational cost, measured by the number of optimisation steps, increases linearly. Lastly, the distance between appAUC and CE minima for the number of optimisation steps, as opposed to Euclidean distance, produces a larger tail in the plot due to the nature of the L-BFGS optimiser. The step size is determined by the L-BFGS formulation, and for a small step size there will be substantially more steps for the same Euclidean distance. We expect the number of steps to grow when the LFL is more locally anharmonic.

### 5.2.2 EXPONENTIAL

The same analysis is done with the exponential surrogate function, $\ell(z) = \exp(-\beta z)$, in Figure 7. Similar conclusions to above are drawn for L-BFGS optimiser steps. However, the number of minima found and the best AUC are found to decrease with increasing $\beta$. Larger values of $\beta$ here cause the exponential function to more closely resemble a step for $z > 0$, but the function doesn't then plateau for $z < 0$ as nicely, which is perhaps the cause of the worse AUCs. It does not make sense to calculate $\Delta$AUC here, as $\exp(-\beta z)$ does not truly attempt to "approximate" the actual step-function AUC. From this we conclude that $\beta = 1$ is the best parameter to use for the exponential function.

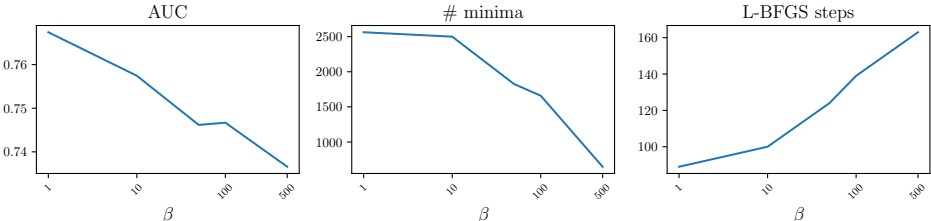

Figure 7: AUC of the best minimum, number of minima of the LFL and number of steps per optimiser basin-hopping run for increasing values of hyperparameter $\beta$ in the Exponential function.

### 5.3 EQUALITY IN EQ.5

The AUC is defined by the following integral:

$$\text{AUC}(\mathbf{W}; \mathbf{X}) = \int_0^1 T(\mathbf{W}; \mathbf{X}, P) \, \mathrm{d}F(\mathbf{W}; \mathbf{X}, P). \tag{10}$$

Substituting in $T$ and $F$ (equations 2 and 3) gives

$$\text{AUC}(\mathbf{W}; \mathbf{X}) = \int_0^1 \frac{\sum_d \delta(c^d - 1)\Theta(p_1(\mathbf{W}; \mathbf{x}^d) - P)}{\sum_d \delta(c^d - 1)} \, \mathrm{d}\left(\frac{\sum_{d'}(1 - \delta(c^{d'} - 1))\Theta(p_1(\mathbf{W}; \mathbf{x}^{d'}) - P)}{\sum_{d'} 1 - \delta(c^{d'} - 1)}\right). \tag{11}$$

We may take outside of the integral any factors that do not depend on the parameter $P$. We note also that the sums $\sum_d \delta(c^d - 1)$ and $\sum_{d'} 1 - \delta(c^{d'} - 1)$ are simply $N_{\text{P}}$ (the number of positive cases) and $N_{\text{N}}$ (the number of negative cases), respectively, and that the sums in the numerator, reduce to taking the sum over all positive points, with $d = \text{p}$, and all negative points, with $d' = \text{n}$:

$$\text{AUC}(\mathbf{W}; \mathbf{X}) = \frac{1}{N_{\text{P}} N_{\text{N}}} \sum_{\text{p}} \sum_{\text{n}} \int_0^1 \Theta(p_1(\mathbf{W}; \mathbf{x}^{\text{p}}) - P) \, \mathrm{d}\Theta(p_1(\mathbf{W}; \mathbf{x}^{\text{n}}) - P). \tag{12}$$

Because the parameter $P$ is continuous, we can write

$$\mathrm{d}\Theta(p_1(\mathbf{W}; \mathbf{x}^{\text{n}}) - P) = -\delta(p_1(\mathbf{W}; \mathbf{x}^{\text{n}}) - P) \, \mathrm{d}P. \tag{13}$$

The limits of the integral must be changed: when $P = 1, \Theta = 0$ and when $P = 0, \Theta = 1$, so $\int_0^1 \rightarrow \int_1^0$ or $-\int_0^1$:

$$\text{AUC}(\mathbf{W}; \mathbf{X}) = \frac{1}{N_P N_N} \sum_p \sum_n \int_0^1 \Theta(p_1(\mathbf{W}; \mathbf{x}^P) - P) \, \delta(p_1(\mathbf{W}; \mathbf{x}^n) - P) \, \mathrm{d}P. \tag{14}$$

Finally, we integrate over this $\delta$-function, using the property $\int f(x)\delta(x - x_0) \, \mathrm{d}x = f(x_0)$, to obtain equation 5:

$$\text{AUC}(\mathbf{W}; \mathbf{X}) = \frac{1}{N_P N_N} \sum_p \sum_n \Theta(p_1(\mathbf{W}; \mathbf{x}^P) - p_1(\mathbf{W}; \mathbf{x}^n)). \tag{15}$$

Note that the minus sign has disappeared, because the integral limits were flipped.

## 5.4 Derivatives

Equation (6) can now be differentiated and treated as a function that can be optimised, just like equation (1). For completeness and comparison, the first and second derivatives of both equations 1 and 6 are given. For compactness, the arguments of each function are omitted, but a superscript is given if the function depends on a specific data point:

$$p_{c^d}(\mathbf{W}; \mathbf{x}^d) \rightarrow p_{c^d}^d, \qquad \sigma(p_1^P - p_1^n) \rightarrow \sigma^{P,n}. \tag{16}$$

Finally, the derivatives are given in component form (e.g. $\partial \text{CE}/\partial \mathbf{W}$ is a vector of length $N_w$, the number of weights), with labels $\mu, \nu...$ denoting the relevant parts of the weight vector, $\mathbf{W}$.

The loss function:

$$\text{CE} = -\frac{1}{N} \sum_{d=1}^N \ln(p_{c^d}^d) + \lambda \mathbf{W}^2 \tag{17}$$

$$\frac{\partial \text{CE}}{\partial w_\mu} = -\frac{1}{N} \sum_{d=1}^N \frac{1}{p_{c^d}^d} \frac{\partial p_{c^d}^d}{\partial w_\mu} + 2\lambda w_\mu \tag{18}$$

$$\frac{\partial^2 \text{CE}}{\partial w_\mu \partial w_\nu} = -\frac{1}{N} \sum_{d=1}^N \left( \frac{1}{p_{c^d}^d} \frac{\partial^2 p_{c^d}^d}{\partial w_\mu \partial w_\nu} - \frac{1}{(p_{c^d}^d)^2} \frac{\partial p_{c^d}^d}{\partial w_\mu} \frac{\partial p_{c^d}^d}{\partial w_\nu} \right) + 2\lambda \delta_{\mu\nu} \tag{19}$$

The approximate AUC function (note an L2 regularisation term has been added, to combat overfitting):

$$A = \frac{1}{N_P N_N} \sum_p \sum_n \sigma^{P,n} + \lambda \mathbf{W}^2 \tag{20}$$

$$\frac{\partial A}{\partial w_\mu} = \frac{1}{N_P N_N} \sum_p \sum_n \left[ \beta \sigma^{P,n}(1 - \sigma^{P,n}) \left( \frac{\partial p_1^P}{\partial w_\mu} - \frac{\partial p_1^n}{\partial w_\mu} \right) \right] + 2\lambda w_\mu \tag{21}$$

$$\frac{\partial^2 A}{\partial w_\mu \partial w_\nu} = \frac{1}{N_P N_N} \sum_p \sum_n \left[ \beta^2 \sigma^{P,n}(1 - \sigma^{P,n})(1 - 2\sigma^{P,n}) \left( \frac{\partial p_1^P}{\partial w_\mu} - \frac{\partial p_1^n}{\partial w_\mu} \right) \left( \frac{\partial p_1^P}{\partial w_\nu} - \frac{\partial p_1^n}{\partial w_\nu} \right) \right.$$
$$\left. + \beta \sigma^{P,n}(1 - \sigma^{P,n}) \left( \frac{\partial^2 p_1^P}{\partial w_\mu \partial w_\nu} - \frac{\partial^2 p_1^n}{\partial w_\mu \partial w_\nu} \right) \right] + 2\lambda \delta_{\mu\nu} \tag{22}$$

Finally, for both the loss function and approximate AUC function, the derivatives of $p_\ell^d$ must be evaluated ($\ell$ is a placeholder label to cover both cases, $c^d$ or 1). Here we separate the four different groups of weights. These groups are: input-hidden weights (denoted as $w^{(2)}$), hidden-output ($w^{(1)}$), bias to hidden nodes ($w^{bh}$) and bias to output nodes ($w^{bo}$).

First derivatives:

$$\frac{\partial p_\ell^d}{\partial w_\mu^{\text{bo}}} = p_\ell^d \Delta_{\ell\mu}^d \tag{23}$$

$$\frac{\partial p_\ell^d}{\partial w_{\mu\nu}^{(1)}} = t_\nu^d \frac{\partial p_\ell^d}{\partial w_\mu^{\text{bo}}} \tag{25}$$

$$\frac{\partial p_\ell^d}{\partial w_\mu^{\text{bh}}} = (s_\mu^d)^2 \sum_{i=1}^C w_{i\mu}^{(1)} \frac{\partial p_\ell^d}{\partial w_i^{\text{bo}}} \tag{24}$$

$$\frac{\partial p_\ell^d}{\partial w_{\mu\nu}^{(2)}} = x_\nu^d \frac{\partial p_\ell^d}{\partial w_\mu^{\text{bh}}} \tag{26}$$

where

$$t_\nu^d \equiv \tanh\left(w_\nu^{\text{bh}} + \sum_{k=1}^{N_{\text{in}}} w_{\nu k}^{(2)} x_k^d\right) \qquad s_\nu^d \equiv \text{sech}\left(w_\nu^{\text{bh}} + \sum_{k=1}^{N_{\text{in}}} w_{\nu k}^{(2)} x_k^d\right) \tag{27}$$

and

$$\Delta_{ij}^d \equiv \delta_{ij} - p_j^d \tag{28}$$

Second derivatives:

$$\frac{\partial^2 p_\ell^d}{\partial w_\mu^{\text{bo}} \partial w_\rho^{\text{bo}}} = p_\ell^d (\Delta_{\ell\rho}^d \Delta_{\ell\mu}^d - p_\mu^d \Delta_{\mu\rho}^d) \tag{29}$$

$$\frac{\partial^2 p_\ell^d}{\partial w_{\mu\nu}^{(1)} \partial w_\rho^{\text{bo}}} = t_\nu^d \frac{\partial^2 p_\ell^d}{\partial w_\mu^{\text{bo}} \partial w_\rho^{\text{bo}}} \tag{33}$$

$$\frac{\partial^2 p_\ell^d}{\partial w_\mu^{\text{bh}} \partial w_\rho^{\text{bo}}} = (s_\mu^d)^2 \sum_{i=1}^C w_{i\mu}^{(1)} \frac{\partial^2 p_\ell^d}{\partial w_i^{\text{bo}} \partial w_\rho^{\text{bo}}} \tag{30}$$

$$\frac{\partial^2 p_\ell^d}{\partial w_{\mu\nu}^{(1)} \partial w_{\rho\sigma}^{(1)}} = t_\nu^d t_\sigma^d \frac{\partial^2 p_\ell^d}{\partial w_\mu^{\text{bo}} \partial w_\rho^{\text{bo}}} \tag{34}$$

$$\frac{\partial^2 p_\ell^d}{\partial w_{\mu\nu}^{(2)} \partial w_\rho^{\text{bo}}} = x_\nu^d \frac{\partial^2 p_\ell^d}{\partial w_\mu^{\text{bh}} \partial w_\rho^{\text{bo}}} \tag{35}$$

$$\frac{\partial^2 p_\ell^d}{\partial w_\mu^{\text{bh}} \partial w_\rho^{\text{bh}}} = \left[(s_\rho^d)^2 \sum_{i,j=1}^C w_{i\mu}^{(1)} w_{j\rho}^{(1)} \frac{\partial^2 p_\ell^d}{\partial w_i^{\text{bo}} \partial w_j^{\text{bo}}}\right.$$

$$\frac{\partial^2 p_\ell^d}{\partial w_{\mu\nu}^{(2)} \partial w_\rho^{\text{bh}}} = x_\nu^d \frac{\partial^2 p_\ell^d}{\partial w_\mu^{\text{bh}} \partial w_\rho^{\text{bh}}} \tag{36}$$

$$\left. - 2\delta_{\mu\rho} t_\mu^d p_\ell^d \sum_{i=1}^C w_{i\mu}^{(1)} \Delta_{\ell i}^d\right] \times (s_\mu^d)^2 \tag{31}$$

$$\frac{\partial^2 p_\ell^d}{\partial w_{\mu\nu}^{(2)} \partial w_{\rho\sigma}^{(1)}} = x_\nu^d \frac{\partial^2 p_\ell^d}{\partial w_{\rho\sigma}^{(1)} \partial w_\mu^{\text{bh}}} \tag{37}$$

$$\frac{\partial^2 p_\ell^d}{\partial w_{\mu\nu}^{(1)} \partial w_\rho^{\text{bh}}} = t_\nu^d \frac{\partial^2 p_\ell^d}{\partial w_\rho^{\text{bh}} \partial w_\mu^{\text{bo}}} + \delta_{\nu\rho} p_\ell^d (s_\rho^d)^2 \Delta_{\ell\mu}^d \tag{32}$$

$$\frac{\partial^2 p_\ell^d}{\partial w_{\mu\nu}^{(2)} \partial w_{\rho\sigma}^{(2)}} = x_\sigma^d \frac{\partial^2 p_\ell^d}{\partial w_{\mu\nu}^{(2)} \partial w_\rho^{\text{bh}}} \tag{38}$$