# OpenReview forum: "Characterising the Area Under the Curve Loss Function Landscape"
_ICLR.cc/2022/Conference — ICLR 2022 Submitted_

### Official Review · Reviewer_ifdg · 2021-11-02

**Correctness:** 3
**Technical Novelty And Significance:** 3
**Empirical Novelty And Significance:** 3
**Recommendation:** 6
**Confidence:** 3

**Main Review:**

Pros
1. authors compared the minimization of CE versus the direct minimization of AUC and its surrogates.
2. authors find a way to visualize the comparisons, which is quite intuitive (Figure 2)
3. authors correlate these losses to demosntrate the relevance between these losses.
4. it is not difficult to understand the effectiveness of surrogates of AUC and the tightness of CE to approximate AUC

Cons
1. the experiment settings are not well addressed. AUC is mostly used for recommendation and advertising. Using such datasets might be more convincing.
2. Some Belkin's work show that there is not so many difference between least square and CE or hinger loss in over-parameterized learning regiem. I am wondering the whether the discussion on the AUC could be included as a special case of their works?

[1] Classification vs regression in overparameterized regimes: Does the loss function matter?
[2] Evaluation of Neural Architectures Trained with Square Loss vs Cross-Entropy in Classification

**Summary Of The Paper:**

Authors did empirical studies to visualize and investigate the landscape of AUC (area under curve) loss. AUC is cirtical for a wide range of applications, such as recommendation and online advertising. Thus it is important.

**Summary Of The Review:**

It is a solid work, though the connections to [1][2] are not clear.

---

### Official Review · Reviewer_Q2LC · 2021-11-03

**Correctness:** 2
**Technical Novelty And Significance:** 2
**Empirical Novelty And Significance:** 2
**Recommendation:** 3
**Confidence:** 3

**Main Review:**

Strengths:
-- The authors have an intriguing idea of analyzing prediction models through their loss landscapes with two different losses (a standard CE loss and a proposed loss that is an approximation of the AUC). This is an interesting idea for a regime to draw conclusions.
-- The experimental setup in 3.4 is interesting, as it suggests that there are learned models with a high CE value but a good AUC that the appAUC-trained models can find. However, this experiment is not studied any further. It would be interesting to explore those models specifically to understand how they can have high CE but have a good AUC, in terms of the effect on actual points. Are these unstable, spiky solutions? Are they overfitting, such that slightly different test sets would yield significantly worse results? This hinted at interesting conclusions but left them under-explored.
Weaknesses:
-- This paper assumes the AUC is a perfect measurement of the ultimate goal of training these models. While acknowledging that the traditional CE loss is only an approximation of the testing metric AUC, it ignores that the testing metric AUC may only be an approximation of the actual goal in our prediction.
-- The datasets considered are too limited. Not only is the main dataset synthetic, but there is only one synthetic dataset and one real dataset. The results are not convincing as a result of the limited experimental setting. The real credit card data is given merely a couple sentences total, meaning the paper is almost exclusively analyzing a single synthetic dataset. This raises many questions, including how it would perform on data of other modalities, as well as whether the results would even replicate on additional synthetic or simple real datasets. The single synthetic dataset is also never shown, leaving the reader unclear about how these two models differ in their learned classifications in any way other than in terms of their AUC.
-- As briefly mentioned in the related work, this optimization is still very similar to standard prediction losses. It throws extra terms in such that an incorrect prediction contributes to making the loss go up in multiple terms, but it is still moving in the same direction. The added terms make it a more complicated, less stable formula. While studying the slightly different landscape is interesting, ultimately it is not practically useful as a loss function.
-- The loss compared to is a simple CE loss, but the data trained on is unbalanced. While the appAUC accounts for the class imbalance, the CE does not. There are very basic balancing techniques that could be used (e.g. proportional sampling) that would be used in practice for the CE model, and I wonder if those would eliminate the minor improvement in test AUC obtained from the appAUC loss. This is just one example of the much bigger problem that the types of models analyzed are far too minimal, with a single CE model compared to the proposed model.
-- Many of the figures are unclear. Despite briefly defining disconnectivity graphs, their implications are not explained. In fact, many conclusions are just summarily stated from informally looking at these graphs (e.g. “the CE landscape is much more funnelled”). This statement is vague and qualitative and its implications are unclear.
-- The correlation between the loss and the true AUC is not especially strong (section 3.3), which is concerning, considering this is the fundamental assumption made in designing the loss.


**Summary Of The Paper:**

The authors propose to minimize approximations of the AUC during training directly, rather than the usual training objectives which are then evaluated with AUC after training.


**Summary Of The Review:**

The model is not convincingly valuable, either as one for study or for practical use. Almost exclusively using a single synthetic dataset with a brief paragraph on a small real dataset is insufficient to draw any conclusions.

---

### Official Review · Reviewer_hg7P · 2021-11-03

**Correctness:** 2
**Technical Novelty And Significance:** 1
**Empirical Novelty And Significance:** 1
**Recommendation:** 3
**Confidence:** 4

**Main Review:**

The paper is well presented and is easy to follow. My main concern about this paper is its contribution. Frankly speaking, from my view, the contribution made in this paper is quite limited, especially given that the approximate AUC is not proposed by the authors. It is true that the authors did observe something from numerical experiments. However, I'm afraid that these observations cannot be articulated as real contributions. In particular, the authors stated that they "provide a theoretical foundation to explain" the observations they made, which, in my view, is overclaimed.

**Summary Of The Paper:**

The paper studied the AUC loss by investigating its approximates and by comparing it with the cross-entropy loss. The landscapes of the optimization based on approximate AUC and based on the cross-entropy loss are also studied. Numerical experiments are conducted.

**Summary Of The Review:**

See above.

---

### Official Review · Reviewer_qWyh · 2021-11-11

**Correctness:** 3
**Technical Novelty And Significance:** 1
**Empirical Novelty And Significance:** 2
**Recommendation:** 3
**Confidence:** 4

**Main Review:**

Writing: In general, this paper is well-written and easy to follow.

Novelty: The significance and novelty of this paper is rather limited for several reasons.
- The approximate surrogate of AUC examined in this paper has actually been proposed in previous research. Given that this paper only gives some empirical validations on an existing method without many innovations in algorithm design or theoretical analysis, I am not sure this work is significant enough to match the ICLR’s standard.
- AppAUC was really an old metric. As far as I am concerned, there are some recently proposed surrogate functions for AUC maximization (see the Relevance part). These methods are not considered in this paper, which largely affects the significance of this work.

Relevance: This paper missed several literatures on AUC maximization which are closely related with this work.
- As commented above, besides appAUC, there are actually many other literatures investigating deep AUC maximization [1,2]. Some of them have also proposed new surrogate function of AUC. These surrogates should be discussed and compared with appAUC.
- As a very relevant research field, online AUC maximization also aims to directly optimize the AUC metric through some computationally efficient surrogate functions for AUC [3]. I would strongly recommend the author to consider this field in the Related Work section.

[1] Large-scale Robust Deep AUC Maximization: A New Surrogate Loss and Empirical Studies on Medical Image Classification.
[2] Stochastic auc maximization with deep neural networks.
[3] Stochastic online AUC maximization.

**Summary Of The Paper:**

This paper studies AUC maximization in classification tasks, which aims to directly optimize the AUC metric instead of minimizing traditional classification losses such as the cross-entropy loss. The main contribution of this paper is an empirical study of an existing AUC maximization method termed approximate AUC. Specifically, this paper compares the landscape of the approximate AUC surrogate loss with that of the cross-entropy loss. The empirical results gives an explanation on why AUC maximization can achieve higher test AUC compared to the conventional method that directly minimizes the cross-entropy loss.

**Summary Of The Review:**

Overall, the novelty of this paper is rather limited and the relevance with previous work is not clear enough, hence I would vote for rejection. I would encourage the authors to consider more recent research on deep AUC maximization, and possibly elaborate a more systematic empirical study on the new surrogate AUC methods.

---

### Official Review · Reviewer_yYns · 2021-11-17

**Correctness:** 3
**Technical Novelty And Significance:** 3
**Empirical Novelty And Significance:** 2
**Recommendation:** 5
**Confidence:** 2

**Main Review:**

Some approximated AUC loss functions show better performance than standard learning via cross-entropy. However, most cases are about empirical results.
Getting theoretical evidence about the advantages of those loss functions is a promising idea.

However, the evaluation part of the paper is not satisfied standard requirements. First, the paper only considers a simple architecture of a neural network, using one linear layer. It could be irrelevant because the existing models have a much more difficult architecture. Second, just two datasets (synthetic and real ones) are used. Further conclusions could be non-reliable.

Also, it is mentioned that hinge loss is not AUC-consistent, but it is used in the SOTA AUC-M loss that is top-1 in the CheXpert competition.

**Summary Of The Paper:**

The paper compares using the cross-entropy and approximated AUC loss functions in neural network training with AUC evaluation. The authors analyze loss function landscapes via solution space and minima properties.

**Summary Of The Review:**

The authors suggest a useful way to analyze loss function selection, but experiments are not persuasive.
I recommend expanding the evaluation part in terms of the number of datasets and real deep learning models with a sufficient number of layers.

---

### Decision · Program_Chairs · 2022-01-20

**Decision:**

Reject

**Comment:**

The paper analyses the loss landscape induced by AUC loss. Reviewers found critical issues with the paper, and the Authors have not provided feedback. As such I have to recommend rejecting the paper. I thank the Authors for submitting the paper to the ICLR conference. I hope the reviews will be helpful in improving the paper.